# Exploratory Evaluation of Topical Tacrolimus for Prevention of Breast Cancer-Related Arm Lymphedema: A Multicenter Non-Randomized Pilot Study

**DOI:** 10.3390/cancers17233753

**Published:** 2025-11-24

**Authors:** Frederik Gulmark Hansen, Mads Gustaf Jørgensen, Kim Gordon, Christina Kjær, Lena Felicia Carstensen, Mette Tambour, Bibi Gram, Jørn Bo Thomsen, Jens Ahm Sørensen

**Affiliations:** 1Research Unit for Plastic Surgery, Odense University Hospital, 5000 Odense, Denmark; 2Institute of Clinical Research, University of Southern Denmark, 5000 Odense, Denmark; 3Department of Physio- and Occupational Therapy, Lillebelt Hospital, University Hospital of Southern Denmark, 7100 Vejle, Denmark; 4Department of Regional Health Research, University of Southern Denmark, 5000 Odense, Denmark; 5Department of Plastic and Breast Surgery, Lillebaelt Hospital, University Hospital of Southern Denmark, 7100 Vejle, Denmark; 6Department of Physiotherapy and Occupational Therapy, University Hospital of Southern Denmark, 6700 Esbjerg, Denmark; 7Research Unit of Endocrinology, Bariatrics, and Diabetes, University Hospital of Southern Denmark, 6700 Esbjerg, Denmark

**Keywords:** breast cancer, lymphedema, tacrolimus, quality of life, topical therapy, late effects

## Abstract

Breast cancer-related lymphedema is a chronic and often disabling complication that can occur after breast cancer treatment. It causes swelling, discomfort, and reduced quality of life. In this study, we investigated whether a topical immune-suppressing ointment called tacrolimus could help prevent the development of lymphedema. We followed patients who applied tacrolimus daily for one year and compared them to a control group. Although we did not find statistically significant differences, some exploratory patterns in symptom development and quality-of-life measures indicate that further investigation may be worthwhile. These findings provide preliminary insights from a small non-randomized pilot trial and may help inform the design of future research into topical therapies for the prevention of breast cancer-related lymphedema.

## 1. Background

Breast cancer-related lymphedema (BCRL) is caused by disrupted lymphatic flow in breast cancer survivors [1]. The risk of developing BCRL ranges from 5 to 65% depending on treatment type, with axillary lymph node dissection and radiation therapy being the most dominant risk factors [2,3]. It is estimated that 3–5 million patients worldwide suffer from BCRL [2,3] with 80–90% of cases occurring within three years post-treatment [4,5]. BCRL causes swelling, pain, and discomfort in the arm or hand, negatively impacting quality of life (QOL) in patients [6,7]. No pharmacological therapy has yet been shown to prevent BCRL, as highlighted in recent reviews on emerging pharmacologic strategies for lymphedema prevention [8].

The current literature indicates that prospective surveillance and early intervention can mitigate symptoms and is therefore paramount in the management of BCRL [9,10].

CD4+ T cells are increasingly recognized as key mediators in the pathogenesis of lymphedema. Their activation and accumulation contribute to inflammation, fibrosis, and impaired lymphatic function [11,12]. Preclinical studies have shown that the degree of T cell infiltration correlates with the severity of fibrosis and clinical stage of lymphedema [13], and that depletion of T cells can reduce tissue fibrosis and improve lymphatic drainage in mouse models [11,13,14]. As a calcineurin inhibitor, tacrolimus inhibits T-cell activation, and topical application has shown promising results in animal models for reducing inflammation, restoring lymphatic function, and limiting fibrotic changes [12,13,15].

Tacrolimus is currently used for treatment of atopic dermatitis and psoriasis with an established long-term safety profile [16,17,18]. The aim of this pilot trial was to evaluate whether topical tacrolimus ointment could prevent the development or reduce the severity of breast BCRL. We hypothesized that daily application of topical tacrolimus for 12 months following axillary lymph node dissection would reduce the incidence and severity of BCRL compared to a control group that did not receive any intervention.

## 2. Methods

### 2.1. Trial Design and Registration

The trial was conducted from February 2020 to June 2022 as a parallel, multicenter, open-label controlled clinical trial. The patients were evaluated at baseline with follow-up at 3, 6, 9 and 12 months.

We used the Open Patient data Explorative Network (OPEN) to store data in a REDCap database [19]. Prior to inclusion, participants received oral and written information and signed an informed consent form. The trial was registered with the Danish Medicinal Agency (EudraCT no. 2018-003416-50), The Regional Committees on Health Research Ethics of Southern Denmark (S-S-20200094), and at www.ClinicalTrials.gov (NCT04390685), and completed according to the International Council for Harmonisation of Technical Requirements for Pharmaceuticals for Human Use-Good Clinical Practice guidelines, current legislation, and regulatory requirements.

### 2.2. Sample Size Calculation

The original study was designed as a randomized controlled trial powered to detect a 66.6% relative risk reduction in lymphedema incidence. Based on an expected incidence of 30% in the control group and 10% in the intervention group and using a two-sided significance level of 0.05 and a power of 80%, the required sample size was calculated to be 118 participants (59 per group). To account for an anticipated 5% dropout rate at 12 months, the target recruitment was increased to 130 participants (65 per group). Due to recruitment challenges, the study enrolled only 60 participants and was thus completed as a non-randomized pilot study.

### 2.3. Participants

Three hospitals in the Region of Southern Denmark (Odense University Hospital, Lillebelt Hospital, University Hospital of Southern Denmark, Vejle, Denmark and University Hospital of Southern Denmark, Esbjerg, Denmark) recruited patients. The intervention group consisted of patients from Odense University hospital while the control group included patients from Vejle and Esbjerg. Each departments investigators daily reviewed the operation schedule for eligible candidates. Unfortunately, the screening process was not systematically monitored, however, we know that approximately 60 axillary lymph node dissections were performed annually across the three participating hospitals. Based on this, we estimate that around 150 patients were screened during the study period. Due to the informal nature of the screening process, we were unable to report an exact number or construct a complete CONSORT flow diagram.

Eligibility criteria included the following inclusion criteria: female sex; diagnosis of breast cancer; age between 18 and 75 years; planned unilateral axillary lymph node dissection (ALND); ASA score of 1 or 2; ability to provide informed consent; and comprehension of Danish. Exclusion criteria were pregnancy or breastfeeding, a diagnosis of primary lymphedema, and any psychiatric disorder deemed likely to interfere with study participation.

Patients who met the eligibility criteria were contacted 1–4 days prior to their planned ALND (Figure 1). All patients received treatment in accordance with the Danish national guidelines for breast cancer care [20].

### 2.4. Intervention

Participants in the intervention group received 20 tubes of 0.1% tacrolimus (Protopic; Leo Pharma, Ballerup, Denmark) at baseline. The intervention group was instructed to apply a thin layer of tacrolimus ointment to the arm, hand, and axilla once daily for a period of 12 months, starting the day after surgery. At inclusion, instructions were given on how to apply tacrolimus including information about side effects: Burning, stinging, or itching at the application site, redness or swelling at the application site, skin rash or irritation and alcohol intolerance. The control group did not receive any intervention.

### 2.5. Endpoints Measures

The primary outcome was the incidence of lymphedema, defined as a ≥10% increase in arm volume in the at-risk arm compared to baseline, assessed by water displacement volumetry at 12 months. Secondary outcome measures included arm volume measured with water displacement volumetry (WDV), the time to lymphedema diagnosis, bioimpedance spectroscopy (BIS), quality of life (QoL), arm, shoulder, and hand function, lymph flow and function, and adverse events (AEs). Furthermore, participants who either developed lymphedema, defined as ≥10% increase in arm volume, or received a clinical diagnosis by a lymphedema therapist or physician, data on conservative treatments—including manual lymphatic drainage (MLD), compression garment use, and pneumatic compression therapy—were systematically collected at each follow-up. These data were recorded regardless of group allocation.

### 2.6. Baseline Characteristics

The baseline characteristics covered both patient-related (age, height, weight, dominant side, relationship status, smoking status, alcohol consumption) and treatment-related (laterality of surgery, number of lymph nodes removed, and type of oncologic treatment) factors.

### 2.7. Lymphedema Diagnosis

The diagnosis of lymphedema was based on the change in volume between the healthy arm and the arm at risk compared to baseline, measured in percent.
$\;Volume\;difference\;change\;\left(\%\right)=\frac{\;volume\;difference\;\left({follow-up}_{n}\right)}{volume\;difference\;\left(baseline\right)}\times 100$, where n represents the follow up (3, 6, 9, or 12 months). A change of ≥10% was considered diagnostic for lymphedema in this trial. Whenever a patient was diagnosed with lymphedema, they were referred to the lymphedema therapists for bandaging and evaluation of the need for compression sleeves.

The patients’ medical journals were reviewed at all follow-ups to see if they had been diagnosed with lymphedema by a physiotherapist or doctor or had received relevant lymphedema treatment in between follow-ups. This information was collected for both groups and used to support secondary analyses and to ensure relevant follow-up.

### 2.8. Water Displacement Volumetry (WDV)

WDV, a method based on Archimedes’ principle of buoyancy, was used to measure arm volume. The Bravometer (Novuqare BV, PJ Horst, The Netherlands) was used along with the technique described in Damstra et al. [21]. In this method, the patient immerses their arm in a basin filled with water, causing an equal volume of water to be displaced into another basin. The amount of displaced water was then measured in grams and converted to milliliters (1:1). The volumes of both arms were measured at baseline and at the follow-ups.

### 2.9. Patient Reported Outcome Measures (PROMs)

Patient-reported outcome measures (PROMs) were collected using three validated Danish-language questionnaires. Quality of life was assessed with the Lymphedema Functioning, Disability and Health questionnaire (LYMPH-ICF) and the EORTC Breast Cancer-specific Quality of Life Questionnaire module QLQ-BR23 (EORTC QLQ-BR23) [22,23]. Upper limb function was evaluated using the Disabilities of the Arm, Shoulder, and Hand questionnaire (DASH) [24]. The LYMPH-ICF includes domains related to physical function, mental well-being, and participation in daily life for patients with lymphedema. The DASH measures symptoms and functional limitations of the upper extremity. The EORTC QLQ-BR23 contains multiple subscales including arm function, body image, breast symptoms, systemic therapy side effects, sexual functioning, sexual enjoyment, and future perspective.

Participants completed the questionnaires in the outpatient clinic at baseline and at the 3-, 6-, 9-, and 12-month follow-ups. Raw scores from all questionnaires were transformed into standardized scores ranging from 0 to 100, according to each instrument’s scoring manual. For the LYMPH-ICF and DASH questionnaires, higher scores indicated greater symptom burden and functional limitation. For the EORTC QLQ-BR23, higher scores on functional subscales reflected better functioning, while higher scores on symptom subscales indicated more severe symptoms.

### 2.10. Bioimpedance Spectroscopy (BIS)

BIS, performed using SOZO^®^ (ImpediMed, Brisbane, Australia) and the manufacturer’s software(SOZO hub SFT-015), was utilized to measure the impedance of each arm’s extracellular fluid. The device uses electrode pads placed on patients’ hands and feet to assess how the body impede current flow. The measured outcome was the lymphedema index (L-Dex), with L-Dex >10 indicating lymphedema. The following variables were additionally measured: extracellular fluid, intracellular fluid, total body water, fat mass, active tissue mass, extracellular mass, and skeletal muscle mass, expressed as percentages (%) of total body water or body weight, respectively. BIS measurements were performed at baseline and at the 3-, 6-, 9-, and 12-month follow-ups, but only in the intervention group, as the equipment was available exclusively at Odense University Hospital.

### 2.11. Safety

Throughout the project, participants were able to report any adverse events, which was documented in the REDCap database. Specifically, the patients were asked about pain and infection at the application site before being asked an open-ended question regarding any treatment-related complaints at 3, 6, 9, and 12 months.

### 2.12. Statistical Analysis

Data were analyzed using Stata v16 (StataCorp 2021, Stata Statistical Software, Release 16; StataCorp LLC., College Station, TX, USA), with a two-tailed significance level set at *p* < 0.05. Normality of continuous variables was assessed using skewness, kurtosis, and Q–Q plots. The LYMPH-ICF, EORTC QLQ-BR23, and DASH raw scores were transformed into standardized scores ranging from 0 to 100 using each instrument’s official scoring key. Descriptive statistics were used to summarize baseline characteristics. Between-group comparisons of normally distributed variables were performed using the Student’s *t*-test; non-normally distributed data were analyzed using the Mann–Whitney U test. For the mixed-effects models, residual distributions were inspected visually using residual-versus-fitted plots and Q–Q plots. No major violations of model assumptions were identified, and no data transformations were required.

Repeated measures over time were analyzed using mixed-effects linear regression models with a group-by-time interaction term (i.group#i.event) to assess whether changes over time differed between groups. Random effects at the participant level were included to account for within-subject correlation. All available observations were included in the mixed-effects models. Missing data were handled via maximum likelihood estimation, which allows participants with incomplete follow-up to contribute all observed measurements under the missing-at-random assumption. No data were imputed, and all participants with at least one post-baseline assessment were included, resulting in an intention-to-treat–like analytic approach.

Continuous parametric variables are presented as mean ± standard error (SE), and categorical variables as frequency and percentage (%).

## 3. Results

Sixty-one patients were included in this study, 22 patients in the intervention group and 39 patients in the control group (Figure 1). Three patients withdrew from the intervention group, and one patient was lost to follow-up; one changed her mind regarding participation in the trial, one withdrew due to side effects (itching), and one withdrew because her breast cancer was too overwhelming, and she could not foresee participating in the trial. Thus, 18 of 22 patients were treated per protocol in the intervention group. None withdrew from the control; thus 39 patients were eligible for final analysis.

Baseline characteristics are summarized in Table 1 and results are presented in Table 2 and Table 3.

The two groups were of similar age, marital status, employment status, and smoking and alcohol consumption. Body mass index (BMI) differed between the two groups with BMI at 24.38 kg/m^2^ in the intervention group and 27.92 kg/m^2^ in the control group (*p* < 0.01). Surgical treatment also differed between the two groups as the intervention group primarily underwent mastectomies (78.95%) and the control group primarily underwent lumpectomies (69.23%) (*p* < 0.01). More patients in the control group (82.02%) had undergone sentinel node biopsy prior to ALND than in the intervention group (36.84%) (*p* < 0.01). Patients in the intervention group had more lymph nodes removed (22.5 ± 7.01) during ALND than patients in the control group (14.65 ± 6.03) (*p* < 0.001). The number of lymph nodes with metastasis was similar in the two groups, 3.78 ± 3.78 for the intervention group and 3.08 ± 6.05 for the control group (*p* > 0.05). Arm volume difference between the healthy arm and the arm at-risk did not differ between the groups at baseline 22.84 ± 95.69 (intervention) vs. 5.90 ± 115.87 (control) (*p* > 0.05).

### 3.1. Lymphedema Diagnosis

During the 12-month follow-up period, three patients (16.7%) in the intervention group and four patients (10.8%) in the control group met the study-specific lymphedema criterion of a ≥10% volume increase in the at-risk arm compared to baseline (Figure 2). No significant difference was observed between the groups (*p* > 0.05). Among these patients, the mean time to diagnosis was 184.0 days in the intervention group and 162.5 days in the control group (*p* > 0.05). At the 12-month follow-up visit, 7 patients (38.9%) in the intervention group and 13 patients (35.1%) in the control group were diagnosed with lymphedema by either a physiotherapist or a physician, based on documentation in the electronic medical record (*p* > 0.05).

### 3.2. Arm Volume

At 12 months, the mean increase in at-risk arm volume was 80.68 mL (95% CI: 6.8–154.56) in the intervention group and 116.07 mL (95% CI: 64.5–167.65) in the control group, with no significant difference between groups (*p* > 0.05) (Table 2). Similarly, no significant between-group differences were observed at 3, 6, or 9 months.

Within-group (paired) analyses of the intervention group showed that the at-risk arm volume was significantly increased at the 12-month follow-up only (*p* < 0.05), while the at-risk arm volume in the control group was significantly increased at all follow-up points compared to baseline (*p* < 0.05) (Figure 3).

The volume of the healthy arm did not differ significantly between groups at any time point (*p* > 0.05). However, when analyzed within groups, the healthy arm volume was significantly increased at 12 months compared to baseline in both groups.

Lymphedema volume, defined as the volume difference between the at-risk and healthy arms, was also analyzed. At 12 months, the mean increase in lymphedema volume was 95.22 ± 34.94 mL in the intervention group (*p* > 0.05) and 76.26 ± 24.38 mL in the control group. Paired analyses revealed significant increases in lymphedema volume at all time points in the control group (*p* < 0.05), whereas in the intervention group, increases were only significant at 9 and 12 months (*p* < 0.05) (Table 2).

### 3.3. Patient Reported Outcome Measures

No significant differences were observed between the intervention and control groups in PROM scores at any follow-up time point across the three questionnaires (Table 3). When conducting paired analysis in each group separately, the following results were observed:

LYMPH-ICF: Symptom burden increased significantly in the control group across all follow-up points, reaching 10.80 ± 2.37 at 12 months (*p* < 0.05) (Figure 4). In the intervention group, scores reduced significantly at 3 and 6 months (7.17 ± 3.34 and 8.03 ± 3.34, respectively; *p* < 0.05) but returned to near-baseline levels at 9 and 12 months (4.75 ± 3.34 and 5.14 ± 3.40; *p* > 0.05).

DASH: Upper extremity function declined in both groups throughout the study. At 12 months, the DASH score increased from 9.31 ± 1.88 to 16.72 ± 2.43 and from 3.30 ±3.29 to 11.81 ± 3.95 in the intervention group (*p* < 0.05 for both compared to baseline ). (Figure 4).

EORTC QLQ-BR23 (Appendix A): Both groups showed a significant decline in future perspective scores over time, with a mean change of −26.38 ± 4.95 in the control group and −22.32 ± 6.48 in the intervention group at 12 months (*p* < 0.05 for both). Arm symptoms increased significantly in both groups (control: 14.75 ± 3.06; intervention: 11.56 ± 4.00 at 12 months; *p* < 0.05 in both groups). Breast symptom scores also increased significantly in both groups (control: 12.36 ± 3.38; intervention: 11.86 ± 4.42 at 12 months; *p* < 0.05 in both groups).

Systemic therapy side effects increased in the control group from baseline to 12 months (mean change: 9.44 ± 3.21, *p* < 0.05), whereas the intervention group did not show a significant increase (6.80 ± 4.42, *p* > 0.05). No significant changes were observed in body image, sexual functioning, or sexual enjoyment scores in both groups.

### 3.4. Lymphedema Index

The L-Dex index was measured in the intervention group only and increased significantly from −0.11 at baseline to 5.98 ± 2.51 at 12 months (*p* < 0.05) (Table 3). Although this represented a statistically significant increase, the mean L-Dex value remained within the normal range (−10 to +10) and thus did not exceed the threshold typically used to indicate lymphedema.

### 3.5. Adherence to Intervention

Adherence to the intervention varied in the intervention group. One patient applied the ointment more than three times a week but less than seven days a week for nine months, with daily application for the remaining three months. Two patients (11.1%) followed a similar pattern for three months before switching to daily application for the subsequent nine months.

Two patients (11.1%) used the ointment less than three times a week for six months but increased to daily application for the remaining six months.

The remaining patients (72.2%) adhered to daily application with fewer than 10 days off intervention throughout the 12-month follow-up.

### 3.6. Safety (Adverse Events)

AEs were monitored in the intervention group. No serious adverse events related to the intervention were reported during the study period. Six patients (31.58%) experienced AEs: One patient experienced postoperative seroma and an infection in the skin at the operated area. This was managed with antibiotics. One patient (5.26%) reported an itching sensation at the site of ointment application during the first week of intervention. The itching lasted for 7 days and did not recur during the intervention period. Two patients (10.53%) experienced folliculitis, which necessitated a week off the intervention. The folliculitis did not recur. Additionally, two patients (10.53%) experienced alcohol-related flushing. As a result, these patients were advised to abstain from alcohol for the remainder of the intervention period.

## 4. Discussion

This study examined the effect and safety of topical tacrolimus in preventing and mitigating BCRL. At 12 months, no significant differences were observed between the intervention and control groups regarding the primary outcome, lymphedema diagnosis, or secondary outcomes, including arm volume and QOL. Adherence to the intervention was high, with no serious adverse events reported.

The group-by-time analysis revealed differing temporal patterns of arm volume progression. In the control group, at-risk arm volume increased significantly at all follow-up points, while in the intervention group, significant volume increases were delayed until 12 months.

These findings may indicate exploratory patterns suggestive of slower fluid accumulation, although they were not statistically significant and must be considered hypothesis-generating.

However, the groups were not randomized and differed at baseline, including variables such as BMI, surgical approach, and arm volume, which could reflect different underlying risk profiles for lymphedema and limit direct comparability. Statistical adjustment was considered, the small sample size and very low number of lymphedema events rendered multivariable models (e.g., ANCOVA or multivariable regression) statistically unstable and prone to overfitting. We therefore relied on mixed-effects models with a group-by-time interaction, which allow exploration of longitudinal patterns while accounting for individual-level variability. As such, these results should be interpreted cautiously and considered exploratory.

PROMs showed different within-group patterns over time. In the control group, scores from both the LYMPH-ICF and EORTC QLQ-BR23 questionnaires indicated a significant decline in quality of life by 12 months, while in the intervention group, these scores returned to baseline levels after initial deterioration at earlier time points. Both groups experienced a sustained decline in upper extremity function as measured by DASH scores, with no significant differences between them. Although statistically significant between-group differences were not observed in QOL measures, the trends in the intervention group may indicate a more favorable trajectory. These findings are exploratory and should be interpreted with caution, but they raise the possibility that topical tacrolimus could mitigate the impact of BCRL symptoms over time.

Recent reviews suggest a promising potential for pharmacological treatment with anti-inflammatory agents for BCRL [25,26]. Gardenier et al. [15] convincingly showed that topical tacrolimus had preventive properties in a mouse model with induced tail lymphedema. While the rodent tail can mimic the progression of human lymphedema, it is important to note that the pathophysiology may differ significantly. Mouse tails lack lymph nodes and are physiologically different from human limbs [27,28]. We previously investigated the efficacy of topical tacrolimus in 18 patients with established BCRL, observing volume reduction, L-Dex, and improvements in PROMs BCRL [29].

Although the present pilot trial did not demonstrate statistically significant between-group differences, trends in arm volume and QOL trajectories in the intervention group appear consistent with our earlier findings. While the previous study evaluated topical tacrolimus as a treatment for established BCRL, and the current study investigates it as a preventive measure, both suggest that topical tacrolimus may exert beneficial effects on lymphedema development and progression. These findings, taken together, support the rationale for further investigation of topical tacrolimus in larger, more definitive trials.

This study’s strengths include its comprehensive follow-up, a large sample size for a pilot trial, and thorough data collection with minimal missing data, enhancing the reliability of our findings. However, several limitations must be acknowledged. Firstly, the absence of blinding and randomization introduces potential selection and detection biases. Because intervention and control participants were recruited from different hospitals, systematic bias cannot be excluded. Although all centers follow national guidelines, small institutional differences in surgical technique or postoperative care may have influenced outcomes.

The lack of stratification contributed to the observed baseline differences, such as BMI, surgical treatment type, and the number of lymph nodes removed, which may have influenced the lymphedema risk. For instance, higher BMI in the control group may elevate their risk, while more mastectomies and the greater number of lymph nodes and removed in the intervention group elevate their risk. These factors complicate the attribution of outcomes solely to the intervention [7]. Another potential confounder is the degree of engagement in conservative lymphedema management strategies which could have influenced outcomes in the groups.

Even though most participants exhibited high adherence to the intervention, variation in application frequency may have influenced intervention response, complicating the interpretation of efficacy. While adherence was monitored and documented, the heterogeneity in intervention exposure represents a limitation and underscores the importance of standardizing adherence protocols in future trials.

Although the study was originally designed and powered to detect a clinically meaningful difference in lymphedema incidence, only 60 participants were ultimately enrolled. The reduced sample size—less than half of the intended 130 participants—limits the statistical power of the study and increases the risk of type II error, potentially obscuring true intervention effects. This under-enrollment was primarily due to recruitment challenges and disruptions during the COVID-19 pandemic. As a result, the trial was completed as a non-randomized pilot study. While the within-group trends observed in this study are promising, they should be interpreted with caution, given the limited sample size and the study’s exploratory nature. Moreover, there was a large variation in arm volume between patients, which implied that we had less power to detect between group differences than within group differences, where the arm volume of each patient could be considered in the statistical modeling. Between-group differences in risk-arm volume change were small and imprecise. At 12 months, the estimated difference was −35.4 mL (95% CI: −130.8–60.1), indication substantial uncertainty and no evidence of treatment effect.

Post hoc power calculations showed that detecting a reduction in lymphedema incidence of 10% (from 30% to 27%) would require approximately 7108 participants, while a 20% reduction (to 24%) would require 1718 participants, and a 50% reduction (to 15%) would need 242 participants, assuming 80% power and a significance level of 0.05. Given the logistical, financial, and ethical challenges of recruiting and following thousands of participants over time, a study of this scale is unlikely to be feasible. Additionally, the burden on participants and healthcare systems from such a large trial would be substantial, raising questions about cost-effectiveness and the practicality of implementation as well as potential overtreatment of 70% of participants. These estimates highlight the challenge of designing a definitive incidence-based efficacy trial at this early stage. However, smaller mechanistic or high-risk subgroup studies remain both feasible and valuable, particularly those focusing on sensitive intermediate outcomes such as arm volume trajectories, bioimpedance measures, or patient-reported symptoms. Such designs may provide the precision needed to refine effect size estimates and guide the development of a future, adequately powered randomized trial. Besides tacrolimus, ketoprofen has proven to reduce limb volume and epidermal thickness in mice with induced tail lymphedema, as well as decreased inflammation and improved histologic changes in capillary lymphatics [30]. A clinical trial further investigated, and found decreased dermal thickness, collagen deposition, and perivascular inflammation in biopsies, although no significant differences in limb volume or bioimpedance were demonstrated [31]. A retrospective study recently analyzed data on 17 patients with BCRL treated with doxycycline (tetracycline), an antibiotic used to treat patients with chronic filarial secondary lymphedema. They found an increase in quality of life, but no differences in limb volume or bioimpedance [32]. The significant volume-reducing effects observed in animal studies often diminish or disappear when translated to humans, highlighting the need for caution in applying findings from animal models to clinical practice [33].

## 5. Conclusions

This pilot study investigated the potential of topical tacrolimus to prevent breast cancer-related lymphedema. No statistically significant differences were observed between the intervention and control groups. Within-group patterns indicated that while both groups experienced worsening of symptoms and quality of life over time, the intervention group showed a return toward baseline at 12 months. These observations are exploratory and hypothesis-generating. These trends are consistent with earlier findings and preclinical data supporting the anti-inflammatory and lymphatic-modulating effects of tacrolimus.

Based on post hoc power calculations, a large-scale incidence-powered trial may not yet be feasible. However, the findings from this pilot study support the rationale for smaller, targeted, hypothesis-driven studies—particularly those centered on sensitive mechanistic endpoints or high-risk subgroups—to better define the potential clinical effect size and inform the design of a future randomized trial.

## Figures and Tables

**Figure 1 cancers-17-03753-f001:**
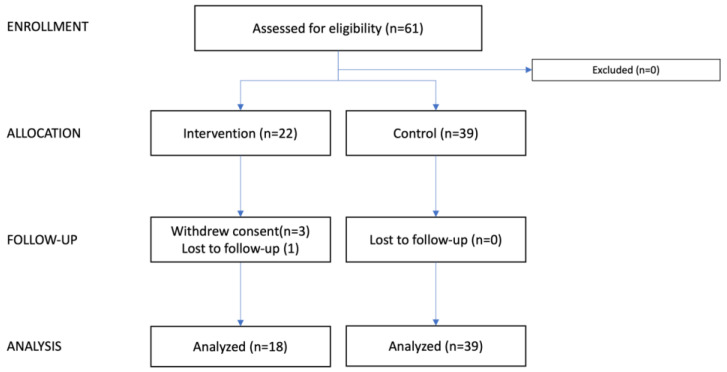
CONSORT flow diagram. This consort flow diagram illustrates the progression of participants through the study. Patients were screened based on the departmental operation schedule; however, as the process was not systematically recorded, the total number of screened patients is unknown. The diagram shows the number of patients assessed, allocated, followed up, and analyzed in both the intervention and control groups.

**Figure 2 cancers-17-03753-f002:**
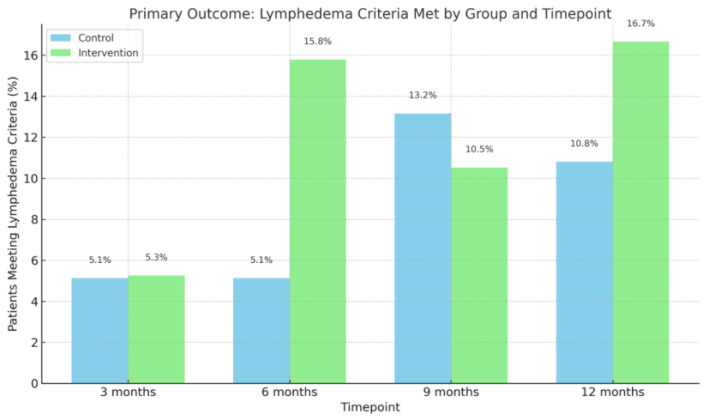
Number of patients meeting lymphedema criteria. Bar chart showing the percentage of patients who met the predefined lymphedema criteria (≥10% volume increase from baseline) in the control and intervention groups at 3, 6, 9, and 12 months. Although proportions varied slightly over time, there were no statistically significant between-group differences at any timepoint.

**Figure 3 cancers-17-03753-f003:**
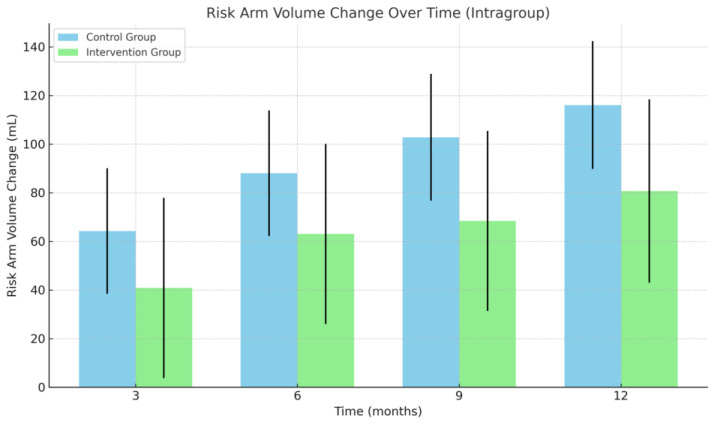
Change in at-risk arm volume. Mean change in risk arm volume over time within each group. Bars represent model-estimated means (coefficients) with corresponding standard errors at 3, 6, 9, and 12 months, based on intragroup estimates from a mixed- effts model. Abbreviations: mL, milliliters.

**Figure 4 cancers-17-03753-f004:**
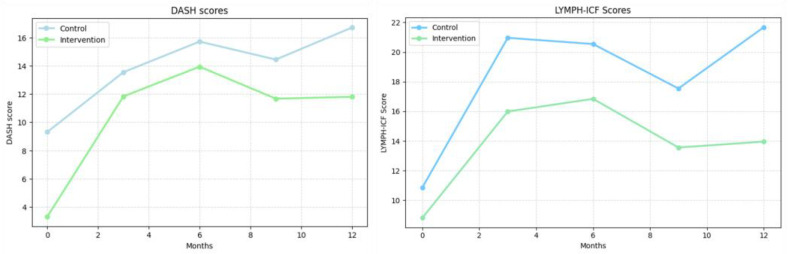
DASH scores (**left**) and LYMPH-ICF scores (**right**). (**Left**): Disabilities of the Arm, Shoulder and Hand (DASH) scores. (**Right**): Lymphedema Functioning, Disability and Health Questionnaire (LYMPH-ICF) scores. Values represent mean scores at each follow-up time point.

**Table 1 cancers-17-03753-t001:** Baseline characteristics.

Variable	Control Group (n = 39)	Intervention Group (n = 22)	*p*-Value
Age (mean ± SD)	53.8 ± 11.43	52.8 ± 9.10	>0.05
In relationship n (%)	20 (71.43%)	12 (66.67%)	>0.05
Smoking n (%)	4 (13.79%)	3 (15.79%)	>0.05
Alcohol consumption (units per week, mean ± SD)	1.61 ± 3.20	2.63 ± 3.53	>0.05
Employment status (working, n (%))	22 (64.71%)	13 (68.42%)	>0.05
Body Mass Index (BMI, mean ± SD)	27.92 ± 5.12	24.38 ± 3.22	**<0.01**
Surgical treatment			**<0.01**
Lumpectomy, n (%)	27 (69.23%)	4 (21.05%)	
Mastectomy, n (%)	10 (30.77%)	15 (78.95%)	
Sentinel node biopsy, n (%)	32 (82.05%)	7 (36.84%)	**<0.01 **
Number of lymph nodes resected (mean ± SD)	14.65 ± 6.03	22.5 ± 7.01	**<0.001**
Number of lymph nodes with metastasis (mean ± SD)	3.08 ± 6.05	3.78 ± 3.78	>0.05
Oncologic treatment, n (%)			
Radiotherapy	36 (92.31%)	18 (100%)	>0.05
Chemotherapy	28 (71.80%)	17 (94.44%)	>0.05
Neo-adjuvant chemotherapy	24 (61.54%)	10 (55.56%)	>0.05
Endocrine therapy	36 (96.31%)	12 (66.67%)	**<0.05**
Biological therapy	4 (10.26%)	2 (12.50%)	>0.05
Zolendronic acid	20 (51.28%)	10 (55.56%)	>0.05
Arm volume difference (mL, mean ± SD)	5.90 ± 115.87	22.84 ± 95.69	>0.05
Operated in dominant side, n (%)	18 (46.15%)	8 (42.11%)	>0.05

Baseline characteristics of the participants in the control and intervention groups. Continuous variables are presented as mean ± standard deviation (SD), and categorical variables are shown as counts with percentages. Statistically significant *p*-values (*p* < 0.05) are written in **bold**. Abbreviations: SD, Standard deviation; n, number.

**Table 2 cancers-17-03753-t002:** Lymphedema diagnosis, volume measurements and time to diagnosis.

Outcome	Control Group	Intervention Group	*p*-Value
Lymphedema diagnosis acording to medical journal (yes)			
3 months, n (%)	2 (5.13%)	1 (5.26%)	>0.05
6 months, n (%)	6 (15.38%)	3 (15.79%)	>0.05
9 months, n (%)	10 (26.32%)	6 (31.58%)	>0.05
12 months, n (%)	13 (35.14%)	7 (38.89%)	>0.05
Lymphedema criteria met? (yes) (≥10% volume change compared to baseline)			
3 months, n (%)	2 (5.13%)	1 (5.26%)	>0.05
6 months, n (%)	2 (5.13%)	3 (15.79%)	>0.05
9 months, n (%)	5 (13.16%)	2 (10.53%)	>0.05
12 months, n (%)	4 (10.81%)	3 (16.67%)	>0.05
Risk arm volume change			
3 months (mean ± SE)	64.26 ± 25.87 mL *	40.84 ± 37.06 mL	>0.05
6 months (mean ± SE)	88.00 ± 25.87 mL *	63.05 ± 37.06 mL	>0.05
9 months (mean ± SE)	102.79 ± 26.09 mL *	68.42 ± 37.06 mL	>0.05
12 months (mean ± SE)	116.07 ± 26.31 mL *	80.68 ± 37.70 mL *	>0.05
Healthy arm volume change			
3 months (mean ± SE)	19.67 ± 20.18 mL	7.21 ± 26.60 mL	>0.05
6 months (mean ± SE)	36.77 ± 20.18 mL	−2.74 ± 26.60 mL	>0.05
9 months (mean ± SE)	31.03 ± 20.35 mL	−6.53 ± 26.60 mL	>0.05
12 months (mean ± SE)	40.23 ± 20.53 mL *	−15 ± 27.05 mL	>0.05
Lymphedema Volume change			
3 months (mean ± SE)	42.03 ± 23.99 mL	−13.74 ± 34.37 mL	>0.05
6 months (mean ± SE)	51.23 ± 23.99 mL *	65.79 ± 34.37 mL	>0.05
9 months (mean ± SE)	71.47 ± 24.18 mL *	74.95 ± 34.37 mL *	>0.05
12 months (mean ± SE)	76.26 ± 24.38 mL *	95.22 ± 34.93 mL *	>0.05
Time to lymphedema diagnosis			
Time (days, mean ± SE)	162.54 ± 26.69 days	184 ± 39.70 days	>0.05

Values represent group means (±SE) at each time point. Asterisks (*) indicate statistically significant within-group changes from baseline (*p* < 0.05). Differences between groups at each time point are represented by *p*-values in the ‘*p*-Value’ column. Abbreviations: SE, Standard Error; mL, milliliters; n, number.

**Table 3 cancers-17-03753-t003:** Outcomes II: L-Dex and patient reported outcome measures.

Outcome	Control Group	Intervention Group	*p*-Value
L-Dex score (Intervention group compared to baseline measurements)			
3 months (mean ± SE)		7.22 ± 2.47 *	<0.01 **
6 months (mean ± SE)		8.75 ± 2.47 *	<0.001 **
9 months (mean ± SE)		9.04 ± 2.47 *	<0.001 **
12 months (mean ± SE)		5.98 ± 2.51 *	<0.05 *
DASH—(0 = best, 100 = worst)			
3 months (mean ± SE)	4.24 ± 1.51 *	8.54 ± 2.15 *	>0.05
6 months (mean ± SE)	6.42 ± 1.50 *	10.65 ± 2.15 *	>0.05
9 months (mean ± SE)	5.14 ± 1.53 *	8.38 ± 2.15 *	>0.05
12 months (mean ± SE)	7.41 ± 1.53 *	8.51 ± 2.18 *	>0.05
LYMPH-ICF—(0 = best, 100 = worst)			
3 months (mean ± SE)	10.09 ± 2.35 *	7.17 ± 33.34 *	>0.05
6 months (mean ± SE)	9.67 ± 2.33 *	8.03 ± 3.34 *	>0.05
9 months (mean ± SE)	6.67 ± 2.39 *	4.75 ± 3.34	>0.05
12 months (mean ± SE)	10.80 ± 2.37 *	5.14 ± 3.40	>0.05
EORTC-BR23—Symptoms: Systemic Therapy Side Effects (0 = best, 100 = worst)			
3 months (mean ± SE)	−0.51 ± 2.92	−4.51 ± 4.15	>0.05
6 months (mean ± SE)	4.32 ± 2.92	4.26 ± 4.15	>0.05
9 months (mean ± SE)	7.09 ± 2.97 *	7.02 ± 4.15	>0.05
12 months (mean ± SE)	9.44 ± 3.21 *	6.80 ± 4.42	>0.05
EORTC-BR23—Symptoms: Upset by Hair Loss (0 = best, 100 = worst)			
3 months (mean ± SE)	16.65 ± 9.74	21.42 ± 17.67	>0.05
6 months (mean ± SE)	14.17 ± 9.84	41.58 ± 25.75	>0.05
9 months (mean ± SE)	9.16 ± 11.54	2.50 ± 18.36	>0.05
12 months (mean ± SE)	35.86 ± 11.70 *	Not answered	N/A
EORTC-BR23—Symptoms: Arm Symptoms (0 = best, 100 = worst)			
3 months (mean ± SE)	9.98 ± 2.77 *	11.11 ± 3.93 *	>0.05
6 months (mean ± SE)	13.30 ± 2.76 *	15.79 ± 3.93 *	>0.05
9 months (mean ± SE)	13.32 ± 2.82 *	15.20 ± 3.93 *	>0.05
12 months (mean ± SE)	14.75 ± 3.06 *	11.56 ± 4.00 *	>0.05
EORTC-BR23—Symptoms: Breast Symptoms (0 = best, 100 = worst)			
3 months (mean ± SE)	17.13 ± 3.06 *	14.47 ± 4.35 *	>0.05
6 months (mean ± SE)	11.41 ± 3.06 *	15.35 ± 4.35 *	>0.05
9 months (mean ± SE)	9.47 ±3.12 *	14.47± 4.35 *	>0.05
12 months (mean ± SE)	12.36 ± 3.38 *	11.86 ± 4.42 *	>0.05
EORTC-BR23—Functional: Body Image (0 = worst, 100 = best)			
3 months (mean ± SE)	8.67 ± 3.38 *	1.75 ± 4.80	>0.05
6 months (mean ± SE)	5.02 ± 3.37	2.63 ± 4.80	>0.05
9 months (mean ± SE)	−2.87 ± 3.44	3.95 ± 4.80	>0.05
12 months (mean ± SE)	−0.11 ± 3.73	−1.26 ± 4.88	>0.05
EORTC-BR23—Functional: Future Perspective (0 = worst, 100 = best)			
3 months (mean ± SE)	−15.09 ± 4.49 *	−15.79 ± 6.37 *	>0.05
6 months (mean ± SE)	−19.16 ± 4.48 *	−22.81 ± 6.37 *	>0.05
9 months (mean ± SE)	−23.17 ± 4.56 *	−24.56 ± 6.37 *	>0.05
12 months (mean ± SE)	−26.38 ± 4.95 *	−22.32 ± 6.48 *	>0.05
EORTC-BR23—Functional: Sexual Functioning (0 = worst, 100 = best)			
3 months (mean ± SE)	2.74 ± 3.38	0.52 ± 4.92	>0.05
6 months (mean ± SE)	−3.41 ± 3.36	0.52 ± 4.92	>0.05
9 months (mean ± SE)	−1.09 ± 3.45	−1.54 ± 4.92	>0.05
12 months (mean ± SE)	−3.39 ± 3.72	0.35 ± 5.12	>0.05
EORTC-BR23: Functional: Sexual Enjoyment (0 = worst, 100 = best)			
3 months (mean ± SE)	2.29 ± 7.05	9.26 ± 7.12	>0.05
6 months (mean ± SE)	7.13 ± 6.49	8.05 ± 7.28	>0.05
9 months (mean ± SE)	12.95 ± 6.56 *	−0.96 ± 6.90	>0.05
12 months (mean ± SE)	12.12 ± 7.06	1.49 ± 7.51	>0.05

Values represent group means (±SE) at each time point. Asterisks (*) indicate statistically significant within-group changes from baseline * (*p* < 0.05) and ** for (*p* < 0.001). Differences between groups at each time point are represented by *p*-values in the ‘*p*-Value’ column. Abbreviations: SE, Standard Error; DASH, Disabilities of the Arm, Shoulder, and Hand; LYMPH-ICF, Lymphedema Functioning, Disability, and Health Questionnaire; L-Dex, Lymphedema Index.

## Data Availability

The datasets generated and/or analyzed during the current study are not publicly available due to data protection regulations and the clinical nature of the data. However, anonymized and cleaned datasets will be stored in a secure institutional repository at Odense University Hospital for 25 years in accordance with Danish legislation and can be shared by the corresponding author upon reasonable request and with appropriate institutional approvals.

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
