# Peer review of "Exploratory Evaluation of Topical Tacrolimus for Prevention of Breast Cancer-Related Arm Lymphedema: A Multicenter Non-Randomized Pilot Study"

_cancers, 2025, doi:10.3390/cancers17233753_

Round 1
Reviewer 1 Report
Comments and Suggestions for Authors
The authors are to be commended for attempting to collect the data for this study. It can be challenging to get all study subjects to comply and report accurately.
There should be a better control group--one that applies a sham cream (vehicle cream, no tacrolimus), in order to get truly meaningful results. Also, a better control group would record dates of any MLD, compression garment wear, and pneumatic compression therapy use.
Lymphedema diagnosis method should be uniform for all subjects--not 10% RVC for some, and then "by either a physiotherapist or a physician, based on documentation in the electronic medical record" (lines 267-275). It is difficult to make this uniform if there are multiple study sites, but presumably all subjects were measured with water immersion.
The Table 1 baseline subject characteristics show that the Control and Intervention groups differed significantly in BMI, Sentinel node biopsy (does this imply worse cancer in the Control group?), Arm volume difference (does this imply worse cancer in the Intervention group?).
Were inflammatory breast cancer/IBC patients included in either Control or Intervention groups? IBC subjects (higher likelihood of lymphedema) could skew results.
There are no L-Dex scores for the Control group in Table 3--in any future studies, these must be collected, so all participating sites must have the same equipment available, calibrated identically.
Another group already published (ref. 14) results for tacrolimus used as a therapeutic for BCRL. The authors of this new manuscript are testing whether tacrolimus can be used as a preventive for BCRL (lines 41-42), so all comparisons should be apples-to-apples (no apples-to-oranges) to the extent possible.
(lines 338-346) Subject intervention adherence differences amount to variable treatment regimens, which confounds data analysis.
The authors do state that the groups were not randomized and differed at baseline, which limits direct comparability (lines 368-369).
The Discussion should address the possibility that the Intervention group was more compliant with MLD, compression garment wear, and/or pneumatic compression therapy use.
This study is a pilot study, so while there can be no re-do, the above robustness concerns should be addressed in any future studies.
Author Response
1) The authors are to be commended for attempting to collect the data for this study. It can be challenging to get all study subjects to comply and report accurately.
Response: We appreciate your kind words and recognition of the challenges involved in collecting high-quality data from study participants. Compliance and accurate self-reporting can indeed be difficult.
2) There should be a better control group--one that applies a sham cream (vehicle cream, no tacrolimus), in order to get truly meaningful results. Also, a better control group would record dates of any MLD, compression garment wear, and pneumatic compression therapy use.
Response: We acknowledge the limitations of the open-label design and the absence of a placebo (vehicle) control. As this was a pilot study, our primary aim was to assess feasibility and detect exploratory trends before launching a more resource-intensive placebo-controlled trial. Regarding conservative therapies, data on MLD, compression garment use, and pneumatic compression were systematically collected every 3 months for all participants who either met the ≥10% volume increase criterion or were clinically diagnosed with lymphedema. This has been clarified in the revised Methods section.
3) Lymphedema diagnosis method should be uniform for all subjects--not 10% RVC for some, and then "by either a physiotherapist or a physician, based on documentation in the electronic medical record" (lines 267-275). It is difficult to make this uniform if there are multiple study sites, but presumably all subjects were measured with water immersion.
Response: Thank you for the comment. All participants were monitored with water displacement volumetry at each visit. The ≥10% volume increase was the predefined primary diagnostic criterion. Clinical diagnoses made by physiotherapists or physicians were only used to support secondary analyses and ensure relevant follow-up. This distinction has now been clarified in the revised Methods section.
4)The Table 1 baseline subject characteristics show that the Control and Intervention groups differed significantly in BMI, Sentinel node biopsy (does this imply worse cancer in the Control group?), Arm volume difference (does this imply worse cancer in the Intervention group?).
Response: Thank you. These baseline differences were already acknowledged and discussed in the manuscript. To further clarify their potential impact, we have elaborated on this in the Discussion section.
5) Were inflammatory breast cancer/IBC patients included in either Control or Intervention groups? IBC subjects (higher likelihood of lymphedema) could skew results.
Response: Thank you for this important point. Inflammatory breast cancer (IBC) was not an exclusion criterion in the study. However, no participants with IBC were enrolled. As such, potential skewing due to inclusion of IBC cases does not apply.
6) There are no L-Dex scores for the Control group in Table 3--in any future studies, these must be collected, so all participating sites must have the same equipment available, calibrated identically.
Response: Thank you for this valid point. L-Dex measurements were only available at one participating center, which is why these data were not collected for the control group. We agree that uniform data collection across sites would have strengthened the study. For this reason, L-Dex was used only for exploratory and secondary analyses, and results based on this measure were interpreted with caution. Future studies should ensure access to standardized equipment across all sites to improve data consistency.
7) Another group already published (ref. 14) results for tacrolimus used as a therapeutic for BCRL. The authors of this new manuscript are testing whether tacrolimus can be used as a preventive for BCRL (lines 41-42), so all comparisons should be apples-to-apples (no apples-to-oranges) to the extent possible.
Response:
Thank you for this important comment. We respectfully clarify that Gardenier et al. (ref. 14) investigated both therapeutic and preventive effects of topical tacrolimus in preclinical lymphedema models. Hence, the scope of their study aligns well with our focus on prevention. Furthermore, we have been careful to distinguish the current study from our own previously published clinical trial (Gulmark Hansen et al., 2022), which evaluated tacrolimus as a treatment for established BCRL. In the present study, we focus exclusively on prevention in high-risk patients. Accordingly, we have revised the manuscript to ensure that any comparisons between studies are limited, appropriate, and clearly contextualized.
8) (lines 338-346) Subject intervention adherence differences amount to variable treatment regimens, which confounds data analysis.
Response: Thank you. We agree and have already acknowledged this limitation in the Discussion, noting that variability in adherence complicates interpretation of treatment efficacy. This has been further elaborated in the discussion section.
9) The authors do state that the groups were not randomized and differed at baseline, which limits direct comparability (lines 368-369).
Response: We thank the reviewer for acknowledging this. As correctly noted, we have explicitly addressed the non-randomized design and baseline differences in the discussion section (lines 368–369), emphasizing the exploratory nature of the results.
10) The Discussion should address the possibility that the Intervention group was more compliant with MLD, compression garment wear, and/or pneumatic compression therapy use.
Response: Thank you for this important point. While we systematically recorded conservative treatment modalities in patients who met criteria for lymphedema, we cannot exclude that differences in engagement with these treatments may have existed between groups prior to diagnosis. We have added this limitation to the discussion.
11) This study is a pilot study, so while there can be no re-do, the above robustness concerns should be addressed in any future studies.
Response: Thank you for acknowledging the exploratory nature of this pilot study. We agree that the robustness concerns you raised should be addressed in future studies, and we are using these insights in planning subsequent trials with more rigorous standardization and control.
Reviewer 2 Report
Comments and Suggestions for Authors
Overall Comments
- This study enables to explore the preventive potential of topical tacrolimus for breast cancer–related lymphedema (BCRL), supported by strong preclinical evidence of its anti-inflammatory and lymphangiogenic effects. The prospective, multicenter design and the use of multiple objective and subjective measures—such as arm volume, bioimpedance spectroscopy, and quality-of-life assessments—enhance the robustness of the findings. Furthermore, the study provides important safety data showing that long-term topical tacrolimus use is well tolerated. Although statistical significance was not achieved, the observed positive trends offer valuable preliminary insights and a strong foundation for future larger-scale randomized trials.
Major comments
- It would be beneficial to classify breast cancer–related lymphedema both before and after the intervention, as this is crucial for accurately assessing the effects of Tacrolimus.
Minor comments
- On Table 1,2, 3, the descriptions say figure legend, which is not a figure and should be in line with scientific writings. Tables usually have a table title (or caption) placed above the table. Sometimes a short explanatory note is added below the table to define abbreviations or clarify data. Also, the figure caption should be written inline, even with this journal format, showing how the figure and its caption are generated.
- The asterisks in the values of all tables indicated as p < 0.05 in the table notes are not clearly explained. Although the note mentions that the comparisons are made relative to the baseline, this should be presented more clearly—preferably generated in a supplementary table included as additional material.
- From the result sections 3.3-3.6, it is better to include a brief summary of the results, showing them in a table or figures, which makes it easier to understand what the pattern looks like.
Author Response
Overall Comments
- This study enables to explore the preventive potential of topical tacrolimus for breast cancer–related lymphedema (BCRL), supported by strong preclinical evidence of its anti-inflammatory and lymphangiogenic effects. The prospective, multicenter design and the use of multiple objective and subjective measures—such as arm volume, bioimpedance spectroscopy, and quality-of-life assessments—enhance the robustness of the findings. Furthermore, the study provides important safety data showing that long-term topical tacrolimus use is well tolerated. Although statistical significance was not achieved, the observed positive trends offer valuable preliminary insights and a strong foundation for future larger-scale randomized trials.
Response: We thank the reviewer for the positive and encouraging feedback. We agree that the observed trends, combined with the robust design and safety profile, support the feasibility of future randomized controlled trials to further investigate the preventive potential of topical tacrolimus for BCRL.
Major comments
- It would be beneficial to classify breast cancer–related lymphedema both before and after the intervention, as this is crucial for accurately assessing the effects of Tacrolimus.
Response: We thank the reviewer for this suggestion. However, as this was a preventive study in a cohort of patients at-risk of lymphedema development, only a minority of patients developed BCRL. Since classification systems such as ISL staging require the presence of lymphedema, applying these uniformly was not feasible or meaningful in the current study design. Nevertheless, we agree that standardized classification of BCRL severity should be incorporated in future trials to better characterize the onset and progression of the condition.
Minor comments
- On Table 1,2, 3, the descriptions say figure legend, which is not a figure and should be in line with scientific writings. Tables usually have a table title (or caption) placed above the table. Sometimes a short explanatory note is added below the table to define abbreviations or clarify data. Also, the figure caption should be written inline, even with this journal format, showing how the figure and its caption are generated.
Response: Thank you for your helpful feedback. We have adjusted the table formatting by revising the titles and moving descriptive information below the tables where appropriate. However, we acknowledge that final layout and styling are determined by the journal’s production team. We are of course happy to adapt further to meet any specific requirements during the typesetting stage.
- The asterisks in the values of all tables indicated as p < 0.05 in the table notes are not clearly explained. Although the note mentions that the comparisons are made relative to the baseline, this should be presented more clearly—preferably generated in a supplementary table included as additional material.
Response: Thank you for this comment. We agree that the use of asterisks should be clearly explained. We have now revised the table footnotes to explicitly state that the asterisks indicate within-group changes from baseline (p < 0.05). However, since these within-group trends are central to the study’s exploratory findings, we believe this information should remain in the main tables and not be moved to supplementary materials.
- From the result sections 3.3-3.6, it is better to include a brief summary of the results, showing them in a table or figures, which makes it easier to understand what the pattern looks like.
Response:
Thank you for your suggestion. We have now added DASH and LYMPH ICF scores as figures in the manuscript, and EORTC scores as supplemental figures to provide a visual summary of the quality-of-life outcomes over time. In addition, we have repositioned Table 3 so it now appears immediately before section 3.3, allowing the table and subsequent textual interpretation to be read in direct continuity. We believe these changes improve the clarity and flow of the results section.
Reviewer 3 Report
Comments and Suggestions for Authors
This manuscript explores the potential preventive effect of topical tacrolimus in reducing breast cancer–related arm lymphedema (BCRL) after axillary lymph node dissection. The topic is clinically relevant and aligns with growing interest in pharmacological strategies to prevent or mitigate lymphedema—a condition with major implications for survivors’ quality of life.
The study is well written, generally clear, and includes comprehensive reporting of methods and results. However, several methodological, statistical, and interpretative limitations need to be addressed before the manuscript is suitable for publication. The authors rightly acknowledge many of these limitations, but some issues require deeper discussion, clarification, or revision to strengthen scientific validity and interpretive caution.
-
Study Design and Validity
-
The non-randomized, open-label design substantially limits causal inference. While this is acknowledged, the title and abstract still imply a degree of causality (“suggest… may delay onset”) that could be toned down to avoid overinterpretation.
-
The baseline differences between groups (BMI, type of surgery, number of lymph nodes removed) introduce significant confounding. The authors should discuss whether statistical adjustment (e.g., regression or ANCOVA) was attempted or feasible to mitigate this bias.
-
-
Sample Size and Power
-
The manuscript explains recruitment limitations, but the post hoc power analysis (lines 418–425) is somewhat confusing and could be clarified. Presenting an effect size estimate (e.g., difference in arm volume or QOL scores) with 95% confidence intervals would better convey the magnitude and uncertainty of observed effects.
-
-
Statistical Analyses
-
Mixed-effects models are appropriate for repeated measures, but it is not entirely clear how missing data were handled. Were all participants included in the longitudinal models (intention-to-treat) or only per-protocol completers?
-
Please specify whether model assumptions (normality, homoscedasticity) were checked and whether any transformations were applied.
-
-
Interpretation of Trends
-
The manuscript repeatedly refers to “trends suggesting delayed onset” without supporting statistical evidence. Given the lack of significant between-group differences, such claims should be reframed as hypothesis-generating rather than suggestive of efficacy.
-
The within-group differences (e.g., delay of volume increase to 12 months in the intervention group) should be explicitly stated as exploratory findings, not evidence of treatment effect.
-
-
Selection Bias and Allocation
-
Participants were recruited from different hospitals depending on group assignment (Odense vs. Vejle/Esbjerg). This introduces systematic selection bias that may affect results. Please expand the discussion on potential institutional differences in surgical technique, postoperative care, or rehabilitation practices.
-
-
Figure and Table Clarity
-
Figure 1 (CONSORT diagram) lacks total screened and excluded counts. Even if estimates are used, the figure should explicitly note that numbers are approximate.
-
Tables 2 and 3 contain many p-values; consider highlighting key comparisons and moving detailed statistics to supplementary materials.
-
-
Clinical Relevance and Future Directions
-
The discussion could better define what effect size or clinical threshold would justify larger trials. Current wording (“several hundred to several thousand participants”) reads overly discouraging; small mechanistic studies might still be feasible and valuable.
- Minor Comments
-
Abstract: Clarify that the study was non-randomized in the first sentence of Methods to avoid confusion.
-
Simple Summary: While informative, it slightly overstates findings—recommend replacing “suggested that tacrolimus might delay onset” with “did not show significant benefit but exploratory signals warrant further study.”
-
Terminology: Use consistent phrasing—sometimes “intervention” and “treatment” are used interchangeably.
-
Typos:
-
Line 58: “controlled multicenter pilot trial was conducted from February 2020 to June 2022” — add a period at the end.
-
Line 406: “elevate their risk” → “may elevate their risk.”
-
-
References: Consider adding a recent 2023–2024 review on pharmacologic prevention of BCRL for completeness (e.g., Breast Cancer Res Treat or Lymphatic Research and Biology).
-
Data Availability: Suggest indicating whether anonymized data could be shared via institutional repository upon request.
-
-
Author Response
This manuscript explores the potential preventive effect of topical tacrolimus in reducing breast cancer–related arm lymphedema (BCRL) after axillary lymph node dissection. The topic is clinically relevant and aligns with growing interest in pharmacological strategies to prevent or mitigate lymphedema—a condition with major implications for survivors’ quality of life.
The study is well written, generally clear, and includes comprehensive reporting of methods and results. However, several methodological, statistical, and interpretative limitations need to be addressed before the manuscript is suitable for publication. The authors rightly acknowledge many of these limitations, but some issues require deeper discussion, clarification, or revision to strengthen scientific validity and interpretive caution.
Response: We thank the reviewer for the positive assessment of the manuscript and for recognizing the clinical relevance and methodological transparency of the study. We also appreciate the reviewer’s constructive comments regarding methodological, statistical, and interpretative aspects. We have addressed each point in detail below and have revised the manuscript accordingly to improve clarity, scientific rigor, and interpretive caution.
-
Study Design and Validity
-
The non-randomized, open-label design substantially limits causal inference. While this is acknowledged, the title and abstract still imply a degree of causality (“suggest… may delay onset”) that could be toned down to avoid overinterpretation.
-
Response: We agree with the reviewer that the non-randomized, open-label design limits causal inference. Although we attempted to use cautious wording, we acknowledge that the phrasing in the title and abstract may still imply causality. We have therefore revised both the title and abstract sections to further reduce causal language and to emphasize the exploratory nature of the findings.
-
-
-
The baseline differences between groups (BMI, type of surgery, number of lymph nodes removed) introduce significant confounding. The authors should discuss whether statistical adjustment (e.g., regression or ANCOVA) was attempted or feasible to mitigate this bias.
-
-
Response: We acknowledge that baseline differences represent important confounders. Although statistical adjustment was considered, the small sample size and very low number of lymphedema events made multivariable models unstable. We therefore relied on mixed-effects models, as prespecified, and have now clarified this limitation and its implications for causal inference in the Discussion.
-
Sample Size and Power
-
The manuscript explains recruitment limitations, but the post hoc power analysis (lines 418–425) is somewhat confusing and could be clarified. Presenting an effect size estimate (e.g., difference in arm volume or QOL scores) with 95% confidence intervals would better convey the magnitude and uncertainty of observed effects.
-
Response: As this was an exploratory pilot trial, a formal a priori power calculation was not feasible due to the absence of reliable estimates for expected effect size, variability, and event rates in this population. Instead, this study aimed to generate preliminary effect estimates to inform sample size calculations for future randomized trials. We furthermore provided effect size estimates with 95% CIs for the main outcome to contextualize the observed trends.
2. Statistical Analyses
-
-
Mixed-effects models are appropriate for repeated measures, but it is not entirely clear how missing data were handled. Were all participants included in the longitudinal models (intention-to-treat) or only per-protocol completers?
-
Response: Thank you for this comment. We have clarified the handling of missing data in the Statistical Analysis section.
-
-
Please specify whether model assumptions (normality, homoscedasticity) were checked and whether any transformations were applied.
-
Response: Thank you for pointing this out. We now clarify that model assumptions were evaluated for all mixed-effects analyses. Residual-versus-fitted plots and Q–Q plots were inspected to assess normality.
-
Interpretation of Trends
-
The manuscript repeatedly refers to “trends suggesting delayed onset” without supporting statistical evidence. Given the lack of significant between-group differences, such claims should be reframed as hypothesis-generating rather than suggestive of efficacy.
-
-
-
The within-group differences (e.g., delay of volume increase to 12 months in the intervention group) should be explicitly stated as exploratory findings, not evidence of treatment effect.
-
- Response: Thank you for this important point. We agree that the wording may imply a degree of efficacy not supported by the statistical findings. We have therefore revised the manuscript to ensure that all descriptions of observed patterns are framed as exploratory and hypothesis-generating, rather than suggestive of treatment effect.
Response:
-
Selection Bias and Allocation
-
Participants were recruited from different hospitals depending on group assignment (Odense vs. Vejle/Esbjerg). This introduces systematic selection bias that may affect results. Please expand the discussion on potential institutional differences in surgical technique, postoperative care, or rehabilitation practices.
-
Response: Thank you for the comment. We agree that center-based allocation introduces potential selection bias. We have added a brief clarification acknowledging that institutional differences may have influenced results.
-
Figure and Table Clarity
-
Figure 1 (CONSORT diagram) lacks total screened and excluded counts. Even if estimates are used, the figure should explicitly note that numbers are approximate.
-
Response: Thank you for this observation. Screening was performed informally across three sites, and exact counts were not recorded. We have added an estimated number of screened patients in the manuscript text (based on annual ALND volume) but have intentionally not inserted approximate figures into the CONSORT diagram to avoid implying false precision. This approach ensures transparency while maintaining adherence to CONSORT reporting principles.
-
-
Tables 2 and 3 contain many p-values; consider highlighting key comparisons and moving detailed statistics to supplementary materials.
-
Response: Thank you for this helpful comment. In revising the tables, we aimed to balance clarity with the detailed reporting requested across all reviewer feedback. We have therefore retained the full p-value information but have now highlighted the most clinically relevant comparisons and improved the table legends for readability. If preferred by the editors, we are happy to move the full statistical output to supplementary material.
Response:
-
Clinical Relevance and Future Directions
-
The discussion could better define what effect size or clinical threshold would justify larger trials. Current wording (“several hundred to several thousand participants”) reads overly discouraging; small mechanistic studies might still be feasible and valuable.
-
Response:
-
- Minor Comments
-
Abstract: Clarify that the study was non-randomized in the first sentence of Methods to avoid confusion.
Response: Corrected accordingly. -
Simple Summary: While informative, it slightly overstates findings—recommend replacing “suggested that tacrolimus might delay onset” with “did not show significant benefit but exploratory signals warrant further study.”
Response: corrected accordingly -
Terminology: Use consistent phrasing—sometimes “intervention” and “treatment” are used interchangeably.
Response: We thank the reviewer for this helpful observation. Throughout the manuscript, terminology has now been standardized: the term “intervention” is used exclusively when referring to the tacrolimus ointment applied within the study protocol, whereas “treatment” is reserved for general lymphedema care or clinical management outside the intervention. We have reviewed the full manuscript to ensure consistency. -
Typos:
-
Line 58: “controlled multicenter pilot trial was conducted from February 2020 to June 2022” — add a period at the end.
Response: Corrected accordingly. -
Line 406: “elevate their risk” → “may elevate their risk.”
Response: Corrected accordingly.
-
-
References: Consider adding a recent 2023–2024 review on pharmacologic prevention of BCRL for completeness (e.g., Breast Cancer Res Treat or Lymphatic Research and Biology).
Response:Thank you for this helpful suggestion. We have now added a recent 2023–2024 review on pharmacologic prevention of BCRL to the Introduction to ensure that the manuscript reflects the most up-to-date evidence in the field. -
Data Availability: Suggest indicating whether anonymized data could be shared via institutional repository upon request.
-
- Minor Comments
Response: Thank you for this suggestion. We have clarified the Data Availability Statement to explicitly indicate that anonymized, cleaned datasets stored in the secure institutional repository at Odense University Hospital can be shared upon reasonable request and with the required institutional approvals.
Round 2
Reviewer 3 Report
Comments and Suggestions for Authors
I thank the authors; the text incorporated most of the suggestions, therefore, in my opinion, it is ready to be published.